

# Genetic recombination and diversity of sapovirus in pediatric patients with acute gastroenteritis in Thailand, 2010–2018

Kattareeya Kumthip[1,2], Pattara Khamrin[1,2], Hiroshi Ushijima[3,4], Limin Chen[5], Shilin Li[5] and Niwat Maneekarn[1,2]

[1] Department of Microbiology, Chiang Mai University, Faculty of Medicine, Chiang Mai, Thailand
[2] Center of Excellence in Emerging and Re-emerging Diarrheal Viruses, Chiang Mai University, Chiang Mai, Thailand
[3] Department of Pathology and Microbiology, Nihon University School of Medicine, Tokyo, Japan
[4] Department of Developmental Medical Sciences, School of International Health, Graduate School of Medicine, University of Tokyo, Tokyo, Japan
[5] Institute of blood transfusion, Chinese Academy of Medical Sciences and Peking Union Medical College, Chengdu, China

## ABSTRACT

**Background.** Human sapovirus (SaV) is an etiologic agent of acute gastroenteritis (AGE) in all age groups worldwide. Genetic recombination of SaV has been reported from many countries. So far, none of SaV recombinant strain has been reported from Thailand. This study examined the genetic recombination and genotype diversity of SaV in children hospitalized with AGE in Chiang Mai, Thailand.

**Methods.** Stool samples were collected from children suffering from diarrhea who admitted to the hospitals in Chiang Mai, Thailand between 2010 and 2018. SaV was detected by RT-PCR and the polymerase and capsid gene sequences were analysed.

**Results.** From a total of 3,057 samples tested, 50 (1.6%) were positive for SaV. Among positive samples, SaV genotype GI.1 was the most predominant genotype (40%; 20/50), followed by GII.1 and GII.5 (each of 16%; 8/50), GI.2 (14%; 7/50), GIV.1 (4%; 2/50), and GI.5 (2%; 1/50). In addition, 4 SaV recombinant strains of GII.1/GII.4 were identified in this study (8%; 4/50).

**Conclusions.** The data revealed the genetic diversity of SaV circulating in children with AGE in Chiang Mai, Thailand during 2010 to 2018 and the intragenogroup SaV recombinant strains were reported for the first time in Thailand.

## INTRODUCTION

Sapovirus (SaV) is one of the important pathogens that cause outbreaks and sporadic cases of acute gastroenteritis (AGE) in people of all ages worldwide (*Torner et al., 2016*). Prevalences of SaV infection have been reported between 0.2% and 39% in children with AGE (*Magwalivha et al., 2018*) and between 2.2% and 15.6% in all age groups (*Oka et al., 2015*; *Varela et al., 2019*). SaV belongs to genus *Sapovirus* of the *Caliciviridae* family. SaV particle is a small, non-enveloped with 30–35 nm in diameter. SaV has a positive-sense, single stranded RNA genome of about 7.1–7.7 kb in length which contains two open

Corresponding author
Niwat Maneekarn,
niwat.m@cmu.ac.th

reading frames (ORFs) (*Green, 2013*; *Oka et al., 2016*). ORF1 encodes for nonstructural proteins (NS1, NS2, NS3, NS4, NS5, and NS6-NS7) and the major capsid protein (VP1). ORF2 encodes for a minor capsid protein (VP2) (*Oka et al., 2015*). Based on the entire nucleotide sequence of VP1 region, SaV can be classified into 19 genogroups (GI-GXIX) and only four genogroups (GI, GII, GIV, and GV) known to infect human (*Yinda et al., 2017*). Human SaV can be further divided into 18 genotypes (GI.1 to GI.7, GII.1 to GII.8, GIV.1, GV.1, and GV.2) (*Kagning Tsinda et al., 2017*; *Liu et al., 2016*; *Oka et al., 2012*; *Oka et al., 2015*; *Xue et al., 2019*) with one additional genotype of GII.NA that has been reported recently (*Diez-Valcarce et al., 2019*). Among these, SaV GI and GII are the most prevalent genogroups detected worldwide while other genogroups have been rarely detected (*Magwalivha et al., 2018*). Like noroviruses, several SaV recombinant strains of both intragenogroup and intergenogroup recombinations have been reported (*Chanit et al., 2009*; *Lasure & Gopalkrishna, 2017*; *Liu et al., 2015*). Within SaV genome, recombination generally occurs in the ORF1, particularly at the junction between the polymerase (NS7) and capsid (VP1) genes (*Hansman et al., 2005*; *Katayama et al., 2004*; *Phan et al., 2006*). Some of SaV recombinant strains have been documented to associate with the outbreaks (*Hansman et al., 2007*; *Lee et al., 2012*). In Thailand, SaV infection in children with diarrhea has been reported previously, however, none of SaV recombinant strain has been reported previously. Therefore, this study aimed to investigate the genetic recombination and genotype diversity of SaV circulating in children hospitalized with AGE in Chiang Mai, Thailand from 2010 to 2018.

## MATERIALS & METHODS

### Sample collection

Stool samples were collected from children who admitted to hospitals with AGE during the period 2010 to 2018 from five major hospitals in Chiang Mai province, northern Thailand, including Maharaj Nakorn Chiang Mai Hospital, Sriphat Medical Center, Nakornping Hospital, Sanpatong Hospital, and Rajavej Chiang Mai Hospital. The age of patients enrolled in this study ranged from neonate to 15 years old. Acute gastroenteritis was defined by watery diarrhea with three or more stool episodes per day for less than 14 days (*Green, 2013*). All specimens were stored at −20 °C until investigation. This work was conducted under the approval of the Research Ethics Committee of the Faculty of Medicine, Chiang Mai University (MIC-2557-02710). The written informed consent form was obtained from parents before samples were collected from their children

### Detection of SaV by RT-PCR

Viral nucleic acid was extracted from 200 µl of the supernatant of a 10% stool suspension prepared in phosphate-buffered saline (pH 7.4) using the Geneaid Viral Nucleic Acid Extraction Kit II (Geneaid, Taiwan) according to the manufacturer's protocol. The viral RNA was reverse transcribed to cDNA using random hexamer primers and RevertAid™ reverse transcriptase (https://www.thermofisher.com/, USA) according to the manufacturer's instruction. SaV was screened by conventional PCR method using GoTaq DNA polymerase (Promega, USA) with primers SLV5731 and SLV5749 (Table 1)
**Table 1  Oligonucleotide primers used for amplification of SaV capsid and polymerase genes.**

| Primer | Sequence (5′–3′) | Direction | Target gene | Position in SaV genome (Accession no: X86560) | Reference |
|--------|------------------|-----------|-------------|-----------------------------------------------|-----------|
| SLV5317 | CTCGCCACCTACRAWGCBTGGTT | Forward | VP1 | 5083–5105 | *Yan et al. (2003)* |
| SLV5749 | CGGRCYTCAAAVSTACCBCCCCA | Reverse | VP1 | 5494–5516 | *Yan et al. (2003)* |
| Sapp36 | GTTGCTGTTGGCATTAACA | Forward | RdRp | 4273–4291 | *Berke et al. (1997)* |
| Sapp128 | GATTACACCAAATGGGATTCCAC | Forward | RdRp | 4354–4376 | *Martinez et al. (2002)* |
| SaV1245R | CCCTCCATYTCAAACACTA | Reverse | RdRp | 5159–5177 | *Harada et al. (2013)* |
| SV-r-c | GCATTGTAGGTGGCGAGAGCC | Reverse | RdRp | 5079–5099 | *Honma et al. (2001)* |

targeting capsid region (VP1 gene) as described previously (*Yan et al., 2003*). The PCR cycling condition was as follows: initial denature at 94 °C for 3 min, 35 cycles step of denature at 94 °C for 1 min, anneal at 58 °C for 1 min, and extend at 72 °C for 1 min, followed by final extension at 72 °C for 10 min. Amplicon size of 434 bp was separated on 1.5% agarose gel electrophoresis and stained with nucleic acid staining solution (RedSafe, INtRON Biotechonology, South Korea) before subjecting to visualize under ultraviolet transilluminator. In addition to SaV screening, the same set of stool specimens were also tested for several other diarrhea-causing viruses, including rotavirus, norovirus, astrovirus, adenovirus, enterovirus, parechovirus, and Aichivirus using the protocol described previously (*Khamrin et al., 2011*).

## Sequence analysis and genotype identification

A Gel/PCR DNA Fragment Extraction Kit (Geneaid, Taiwan) was used to purify amplicons of SaV-capsid gene according to the manufacturer's protocol. All purified PCR products were direct sequenced using Applied Biosystems BigDye® Terminator Cycle Sequencing Kit v3.1 (Life Technologies, USA) with forward and reverse primers SLV5731 and SLV5749, and analyzed by using Applied Biosystems 3100 Genetic Analyzer (Life Technologies, USA). Nucleotide sequences of partial capsid gene were analyzed manually using BioEdit and Clustal X softwares. Identification of virus genotype was initially determined by using the BLAST Tool (https://blast.ncbi.nlm.nih.gov/Blast.cgi) and Human Calicivirus Typing Tool (https://norovirus.ng.philab.cdc.gov/) and confirmed by phylogenetic analysis using MEGA7 software package (*Kumar, Stecher & Tamura, 2016*). The tree was statistically supported by bootstrapping with 1,000 replicates.

## Recombination analysis

To examine genetic recombination of SaV strains detected in this study, the polymerase (RdRp) region was amplified by using primers shown in Table 1. The PCR amplification was carried out under the same condition as used for the VP1 amplification except for the annealing step which was performed at 45 °C for 1 min. Amplicon sizes of the amplified RdRp region were varied depending on the used primers (746 bp, 824 bp, 827 bp, or 905 bp) shown in Table 1 (*Berke et al., 1997*; *Harada et al., 2013*; *Honma et al., 2001*; *Martinez et al., 2002*). Phylogenetic tree of the partial capsid gene sequences was constructed by using the Maximum Likelihood method based on the Kimura 2-parameter model. Phylogenetic
tree of the partial RdRp region was inferred by using the Maximum Likelihood method based on the General Time Reversible model. To predict the putative recombination point for SaV recombinant strains, nucleotide sequences of the polymerase and capsid genes, spanning the RdRp-VP1 junction region (positions 5078-5265 according to SaV genome of accession no. X86560), were analyzed. The possible recombination point of SaV strain was determined by using SimPlot software v.3.5.1 (*Lole et al., 1999*). In addition, the confidence interval for recombination between the query strain and parent strains was examined by the Recombination Detection Program v.4.71 (RDP4) complemented with the Max-Chi test to confirm the significant events ($p < 0.01$) (*Martin et al., 2015*).

### Nucleotide accession number

Nucleotide sequences of the polymerase and capsid regions of SaV strains detected in this study were deposited in the GenBank database under the accession numbers MN245671 to MN245579 and MN253492 to MN253541 for the polymerase and capsid gene sequences, respectively.

## RESULTS

### Prevalence of SaV

A total of 3,057 stool samples collected between 2010 and 2018 were screened for the presence of SaV. The average of SaV detection rate over the study period of nine years was 1.6% (50 out of 3057) (Table 1). From 2010 to 2015, SaV was detected at low prevalence (0-1.8%). After that, the prevalence of SaV infection increased year by year to 2.2%, 2.5%, and 2.8% in 2016, 2017, and 2018, respectively. There was no specific seasonal pattern of SaV detection observed in this study. The age of SaV-infected patients varied from 8 months up to 11 years (Table 2). Among 50 SaV positive cases, 26 (52%) were male and 24 (48%) were female. All SaV positive samples were also tested for the presence of other enteric viral pathogens. The results showed that majority of cases were single infection with SaV (74%, 37 out of 50) whereas the rest of cases (26%, 13 out of 50) were co-infected with other enteric viruses including rotavirus, norovirus, adenovirus, parechovirus, enterovirus, or astrovirus (Table 3).

### Molecular characterization of SaV and phylogenetic analysis

The partial capsid gene sequence (308 nucleotides) of 50 SaV strains were analyzed to identify genotype of the virus. Seven different genotypes were identified in the present study: GI.1 (20), GI.2 (7), GI.5 (1), GII.1 (8), GII.4 (4), GII.5 (8), and GIV.1 (2). Phylogenetic analysis revealed that 20 SaV GI.1 strains were clustered with the GI.1 strains reported previously from USA, Korea, Japan, and Brazil (Fig. 1) and showed 97.4–99.6% nucleotide homology to theses strains. Seven SaV GI.2 strains were grouped together with the GI.2 strains reported previously from Brazil and USA with highest nucleotide sequence similarity (98.7–100%). GI.5 strain was closely related to the GI.5 Ehime643 strain detected previously in Japan (99.0% nucleotide sequence identity). Eight GII.1 strains shared high similarity (90.3–95.0% nucleotide sequence identities) with GII.1 strains detected previously in United Kingdom, USA, India, Japan, and Thailand. Four GII.4 strains were closely related

**Table 2** Prevalence and genotype distribution of sapovirus detected in children with acute gastroenteritis in Chiang Mai, Thailand from 2010 to 2018.

| Years | Total samples | Positive samples (%) | SaV genotypes | | | | | | |
|-------|---------------|----------------------|------|------|------|-------|-------|-------|-----------|
| | | | GI.1 | GI.2 | GI.5 | GII.1 | GII.5 | GIV.1 | GII.1/GII.4 |
| 2010 | 109 | 2 (1.8) | 2 | – | – | – | – | – | – |
| 2011 | 302 | 0 | – | – | – | – | – | – | – |
| 2012 | 341 | 2 (0.6) | – | – | – | – | – | 1 | 1 |
| 2013 | 280 | 4 (1.4) | 2 | – | – | 1 | – | – | 1 |
| 2014 | 268 | 0 | – | – | – | – | – | – | – |
| 2015 | 335 | 6 (1.8) | 3 | 3 | – | – | – | – | – |
| 2016 | 508 | 11 (2.2) | 6 | 3 | – | – | 2 | – | – |
| 2017 | 278 | 7 (2.5) | 3 | – | – | 2 | – | – | 2 |
| 2018 | 636 | 18 (2.8) | 4 | 1 | 1 | 5 | 6 | 1 | – |
| 9 years | 3057 | 50 (1.6) | 20 | 7 | 1 | 8 | 8 | 2 | 4 |

to GII.4 strains reported from Philippines and Peru (94.5–97.1% nucleotide identities). All GII.5 strains shared highest similarity (90.3–95.0%) with the GII.5 SataRosa3693/2015/GT strain detected in Guatemala. The CMH-N061-12 SaV GIV.1 strain displayed 97.9–99.6% nucleotide identities to GIV.1 strains reported previously from USA and Japan while CMH-N028-18 SaV GIV.1 strain shared 87.1–87.8% sequence homology with those of the SaV GIV.1 reference strains.

## Recombination analysis

In the SaV genome, recombination event typically occurs between the polymerase (RdRp) and capsid (VP1) genes. To investigate the genetic recombination of SaV strains detected in this study, we further amplified the polymerase gene of the viruses. The partial RdRp gene sequence (745 nucleotides) was successfully obtained from 49 of 50 SaV-positive samples. Phylogenetic tree of the partial RpRp region of 49 SaV strains was constructed (Fig. 2) and the results showed that all strains, except for four GII.4 strains (CMH-S057-17, CMH-S050-17, CMH-N021-13, and CMH-N145-12), were clustered into the same genotypes assigned by the capsid gene sequence as shown previously in Fig. 1. Based in the capsid gene sequence (Fig. 1), CMH-S057-17, CMH-S050-17, CMH-N021-13, and CMH-N145-12 were assigned as the GII.4 genotype whereas based on the RdRp sequence they clustured together with SaV GII.1 reference strains and shared 91.8–95.0% nucleotide sequence identities. The data suggested that these four SaV strains were the SaV GII.1/GII.4 recombinant strains. To predict the putative recombination point for SaV recombinant strains, SimPlot software was used. The similarity plot of SaV recombinant strains detected in this study are shown in Fig. 3. Two recombination break points were identified at positions 5088 and 5091 within the RdRp-VP1 junction region (nucleotide positions 5078-5265). The CMH-145-12 and CMH-N021-13 strains showed the same recombination point at position 5088 ($p = 1.572 \times 10^{-14}$ and $p = 1.496 \times 10^{-12}$, respectively) while other 2 strains (CMH-S050-17 and CMH-S057-17) displayed the recombination point at position 5091 ($p = 2.334 \times 10^{-14}$ and $p = 2.322 \times 10^{-14}$, respectively).

**Table 3  General information of patients and genotypes of sapovirus based on polymerase and capsid nucleotide sequences.**

| Sample ID | Collection date | Age | Gender | RdRp genotype | VP1 genotype | Mix-infection |
|---|---|---|---|---|---|---|
| CMH-N018-10 | 12-Dec-2010 | – | Female | GI.1 | GI.1 | – |
| CMH-S050-10 | 14-Dec-2010 | – | Female | GI.1 | GI.1 | – |
| CMH-N061-12 | 11-Feb-2012 | – | Female | GIV.1 | GIV.1 | – |
| CMH-N145-12 | 10-Aug-2012 | – | Male | GII.1 | GII.4 | – |
| CMH-S004-13 | Jan-2013 | – | Male | GI.1 | GI.1 | – |
| CMH-S034-13 | Apr-2013 | – | Female | GII.1 | GII.1 | Rotavirus |
| CMH-N021-13 | 02-May-2013 | – | Male | GII.1 | GII.4 | Rotavirus |
| CMH-N131-13 | 03-Oct-2013 | – | Female | GI.1 | GI.1 | – |
| CMH-S152-15 | 16-Jun-2015 | 28 months | Male | GI.2 | GI.2 | – |
| CMH-S166-15 | 13-Aug-2015 | 35 months | Female | GI.1 | GI.1 | – |
| CMH-S252-15 | 20-Dec-2015 | 54 months | Female | GI.2 | GI.2 | – |
| CMH-S254-15 | 31-Dec-2015 | 32 months | Female | GI.2 | GI.2 | Norovirus |
| CMH-N006-15 | 09 -Jul-2015 | 7 months | Male | GI.1 | GI.1 | Adenovirus |
| CMH-N007-15 | 09 -Jul-2015 | 9 months | Male | GI.1 | GI.1 | Parechovirus |
| CMH-S108-16 | 29-Mar-2016 | 20 months | Female | GI.2 | GI.2 | – |
| CMH-S198-16 | 11-Aug-2016 | 32 months | Male | GI.1 | GI.1 | Enterovirus |
| CMH-S229-16 | 12-Sep-2016 | 20 months | Male | GI.1 | GI.1 | – |
| CMH-ST004-16 | 11-Jan-2016 | 27 months | Female | GI.2 | GI.2 | – |
| CMH-ST016-16 | 02-Feb-2016 | 22 months | Male | GI.1 | GI.1 | – |
| CMH-ST029-16 | 07-Feb-2016 | 143 months | Female | GI.2 | GI.2 | – |
| CMH-ST090-16 | 21-Mar-2016 | 26 months | Female | GI.1 | GI.1 | Norovirus |
| CMH-ST091-16 | 21-Mar-2016 | 8 months | Female | GII.5 | GII.5 | – |
| CMH-ST095-16 | 23-Mar-2016 | 8 months | Female | GII.5 | GII.5 | – |
| CMH-ST163-16 | 13-Aug-2016 | 24 months | Female | GI.1 | GI.1 | Parechovirus |
| CMH-ST199-16 | 12-Dec-2016 | 27 months | Male | GI.1 | GI.1 | Parechovirus |
| CMH-S003-17 | 22-Feb-2017 | 43 months | Male | GI.1 | GI.1 | Astrovirus |
| CMH-S023-17 | 04-Apr-2017 | 34 months | Male | GI.1 | GI.1 | – |
| CMH-S050-17 | 16-Jun-2017 | 16 months | Female | GII.1 | GII.4 | – |
| CMH-S057-17 | 21-Jun-2017 | 22 months | Female | GII.1 | GII.4 | – |
| CMH-S089-17 | 21-Jun-2017 | 12 months | Female | GI.1 | GI.1 | Enterovirus |
| CMH-S120-17 | 21-Nov-2017 | 16 months | Male | GII.1 | GII.1 | – |
| CMH-ST028-17 | 09-May-2017 | 24 months | Male | GII.1 | GII.1 | – |
| CMH-N028-18 | 02-Feb-2018 | 16 months | Female | GIV.1 | GIV.1 | Rotavirus, Parechovirus |
| CMH-N061-18 | 18-Feb-2018 | 26 months | Female | GI.5 | GI.5 | Rotavirus |
| CMH-N091-18 | 24-Jun-2018 | 18 months | Male | GII.1 | GII.1 | – |
| CMH-N104-18 | 10-Sep-2018 | 11 months | Male | GII.1 | GII.1 | – |
| CMH-R031-18 | 21-Oct-2018 | 24 months | Female | GI.1 | GI.1 | – |
| CMH-R076-18 | 21-Oct-2018 | 24 months | Male | GI.2 | GI.2 | – |
| CMH-R089-18 | 21-Oct-2018 | 43 months | Male | GII.5 | GII.5 | – |
| CMH-R140-18 | 21-Oct-2018 | 12 months | Female | GII.1 | GII.1 | – |
| CMH-S174-18 | 21-Nov-2018 | 48 months | Male | GI.1 | GI.1 | – |

**Table 3** (*continued*)

| Sample ID | Collection date | Age | Gender | RdRp genotype | VP1 genotype | Mix-infection |
|---|---|---|---|---|---|---|
| CMH-S175-18 | 21-Nov-2018 | 31 months | Male | GII.1 | GII.1 | – |
| CMH-ST097-18 | 16-Apr-2018 | 25 months | Male | GI.1 | GI.1 | – |
| CMH-ST169-18 | 27-Jun-2018 | 12 months | Male | GI.1 | GI.1 | – |
| CMH-ST189-18 | 27-Jul-2018 | 12 months | Male | GII.1 | GII.1 | – |
| CMH-ST202-18 | 15-Aug-2018 | 20 months | Male | GII.5 | GII.5 | – |
| CMH-ST207-18 | 17-Aug-2018 | 22 months | Male | GII.5 | GII.5 | – |
| CMH-ST247-18 | 10-Nov-2018 | 44 months | Male | GII.5 | GII.5 | – |
| CMH-ST266-18 | 15-Dec-2018 | 24 months | Female | – | GII.5 | – |
| CMH-ST270-18 | 29-Dec-2018 | 32 months | Female | GII.5 | GII.5 | – |

**Notes.**
The highlighted samples are the ones in which recombinant sapoviruses were found.

## SaV infection in different age groups of patients

Among SaV positive cases, the highest detection rate was seen in patients with the age of 2 to <3 years (40.5%), followed by 1 to <2 years (35.7%), less than 1 year (9.5%), 3 to <4 years (7.1%), 4 to <5 years (4.8%), and more than 5 years (2.4%). In addition, this study identified 6 different genotypes and one recombinant pattern of SaV. Distribution of SaV genotypes detected in different age groups of patients is shown in Fig. 4. It was found that GI.1 genotype was detected in all age groups except for patients with more than 5 years of age. The GII.5 was also identified in patients with age groups of less than 4 years. Interestingly, recombinant SaV GII.1/GII.4 was detected in patients between 1 and 2 years of age.

## DISCUSSION

Enteric caliciviruses including noroviruses and sapoviruses are the major causes of AGE of human in all age group worldwide (*Khamrin et al., 2017*; *Lu et al., 2014*; *Neo et al., 2017*; *Sala et al., 2014*). The detection rate of SaV infection in many countries around the world have been reported with a range from 0.2 to 39% in children and 2.2 to 15.6% in all age groups (*Magwalivha et al., 2018*; *Oka et al., 2015*; *Varela et al., 2019*). In Thailand, the prevalence of SaV was detected between 0% and 15.0% (*Kumthip & Khamrin, 2018*). The overall SaV-positive rate reported in the present study during the period 2010 to 2018 (1.6%, ranged from 0 to 2.8%) is more or less the same as other similar studies including Bangladesh (2.7%) (*Dey et al., 2007*), China (1.9%) (*Ren et al., 2013*), Vietnam (1.4%) (*Trang et al., 2012*), and Thailand (1.2 and 1.9%) (*Khamrin et al., 2017*; *Supadej et al., 2019*). However, when compared to the studies conducted in different geographical regions, the prevalence of SaV detected in this study was lower than those reported from Japan (4.8%), Philippines (7%), USA (5.4%), Italy (6%), Denmark (8.8%), Finland (9.3%), Spain (15.6%), and Nicaragua (17%) (*Biscaro et al., 2018*; *Bucardo et al., 2014*; *Chhabra et al., 2013*; *Oka et al., 2015*; *Thongprachum et al., 2015*; *Varela et al., 2019*). The variation of SaV prevalence in different studies could be explained, at least in part, by the difference in study locations, the detection methods, and the emergence of new epidemic strains. It was noticed that our study as well as other studies that had similar SaV prevalence used single

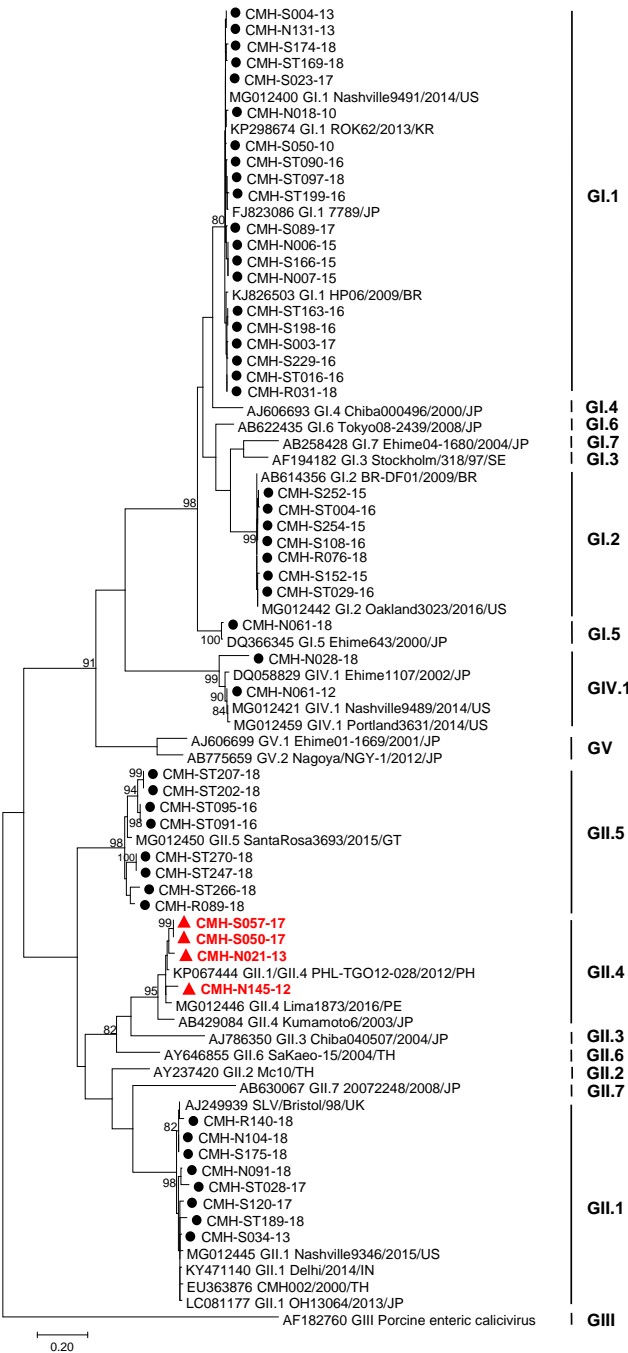

**Figure 1** **Phylogenetic tree of the partial capsid gene sequences (296 nucleotides).** Fifty SaV strains detected in this study are indicated by black circle (non-recombinant strains) and red triangle (recombinant strains). The evolutionary history was inferred by using the Maximum Likelihood method based on the Kimura 2-parameter model. Scale bar indicates nucleotide substitutions per site and bootstrap values (>80) are indicated for the corresponding nodes.

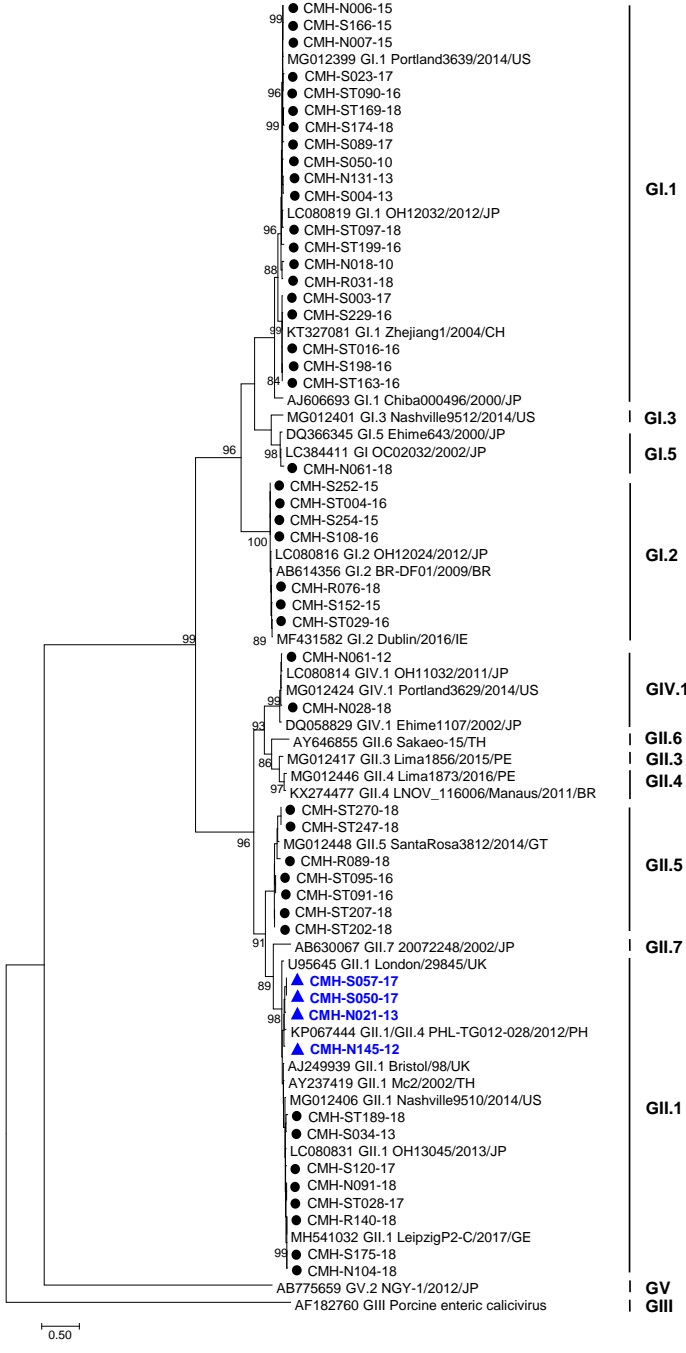

**Figure 2 Phylogenetic tree of the partial RdRp gene sequences (745 nucleotides).** Forty-nine SaV strains detected in this study are indicated by black circle (non-recombinant strains) and blue triangle (recombinant strains). The evolutionary history was inferred by using the Maximum Likelihood method based on the General Time Reversible model. Scale bar indicates nucleotide substitutions per site and bootstrap values (>80) are indicated for the corresponding nodes.

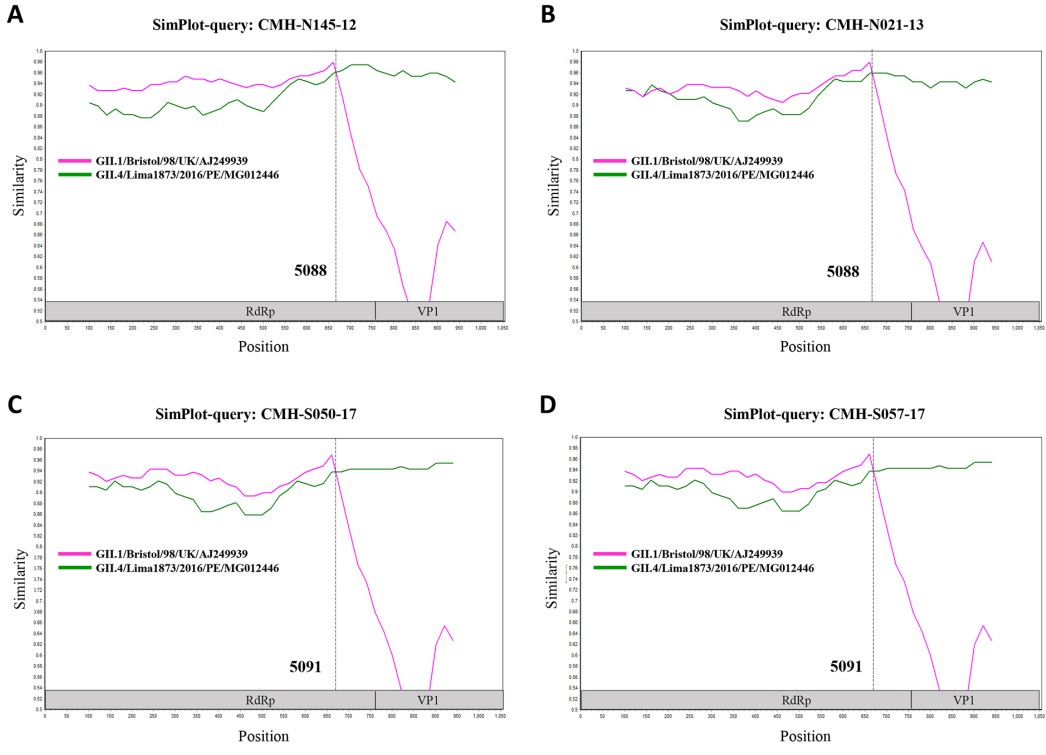

**Figure 3** **The similarity plot of four SaV recombinant strains, CMH-N145-12 (A), CMH-N021-13 (B), CMH-S050-17 (C), CMH-S057-17 (D), was constructed using SimPlot software.** The similarity score of 200 nucleotides sliding window and 20 site step were used. The junction of RdRp and capsid sequences of SaV recombinant strains (1,055 nucleotides) were plotted against the nucleotide sequences of the reference strains, GII.1/Bistol/98/UK and GII.4/Lima1873/2016/PE. The *x*-axis represents the nucleotide position and the *y*-axis indicates the similarity between the query strain and the reference strains.

round PCR as a screening method while other studies that reported higher SaV-positive rate performed real-time PCR for the screening process.

Among different genogroups of SaV, the SaV GI (GI.1 and GI.2) and GII (GII.1 and GII.2) are the most predominant genogroups circulating worldwide while other genotypes are rarely detected in some particular countries (*Diez-Valcarce et al., 2018*; *Magwalivha et al., 2018*). Similar to other reports, SaV GI.1, GI.2, and GII.1 were the most common genotypes detected in our study. Nonetheless, there was no GII.2 strain observed in the present study. The SaV GII.5 is not often detected. The occurrence of GII.5 in human stool samples was reported in some particular areas such as Guatemala, Peru, South Africa, and USA (*Diez-Valcarce et al., 2018*; *Liu et al., 2016*; *Murray et al., 2016*). In addition to these countries, it should be noted that a high proportion of GII.5 strain (16%, 8 out of 50) was observed in our study. Interestingly, 7 out of 8 SaV GII.5-infected patients were from the same hospital (Sanpatong hospital) and 5 of them were detected in the same year of 2018, suggesting that this particular genotype is circulating in a particular location.

RNA recombination plays a key role in virus evolution and leads to its pathogenicity and virus diversity (*Worobey & Holmes, 1999*). At least, two types of recombination events of
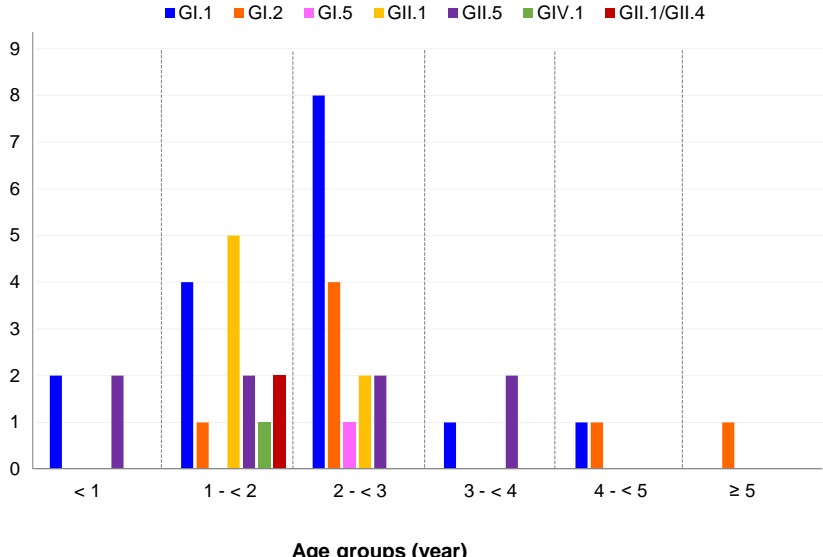

Number of patients

Figure 4 **Distribution of SaV genotypes detected in different age groups of patients.**

SaV including intergenogroup and intragenogroup have been reported previously (*Chanit et al., 2009*; *Hansman et al., 2007*; *Hansman et al., 2005*; *Katayama et al., 2004*; *Oka et al., 2015*; *Phan et al., 2006*). In the present study, intragenogroup recombinant GII.1/GII.4 SaV was detected in four samples, accounting for 8% (4/50) of all SaV infected cases. The recombinant GII.1/GII.4 SaV has been reported previously in Philippines, Vietnam, and USA (*Diez-Valcarce et al., 2018*; *Liu et al., 2015*; *Nguyen et al., 2008*). However, it has not been described elsewhere in Thailand. To our knowledge, this is the first report demonstrating the presence of recombinant SaV in Thailand. Generally, characterization of SaV is based on the nucleotide sequence of capsid gene (*Oka et al., 2012*) and many other previous studies have identified SaV genotypes based only on this gene. To date, both capsid and polymerase genes are used to classify genotype of noroviruses (*Chhabra et al., 2019*). Since SaV is a very similar pathogen in many aspects, therefore, future classification and characterization of SaV should rely on both polymerase and capsid sequences to identify the virus diversity. In addition, continued surveillance on SaV is important to monitor the emergence of new virus strains.

## CONCLUSIONS

In summary, the results of this study highlight the impact of SaV in diarrheal diseases among children in Chiang Mai, Thailand over the period of nine years and is the first report to describe recombinant SaV infection in Thai children suffering with AGE. The data of nucleotide analysis of both polymerase and capsid genes from this study provide useful information for a better understanding on the caliciviruses other than noroviruses.

## ACKNOWLEDGEMENTS

We are grateful to the physicians and nursing staff who helped to collect the specimens used in this study.

### Funding

This work was supported by the Center of Excellence (Emerging and Re-emerging Diarrheal Viruses Research), Chiang Mai University, Chiang Mai, Thailand. The funders had no role in study design, data collection and analysis, decision to publish, or preparation of the manuscript.

### Grant Disclosures

The following grant information was disclosed by the authors:
Center of Excellence (Emerging and Re-emerging Diarrheal Viruses Research), Chiang Mai University, Chiang Mai, Thailand.

### Competing Interests

The authors declare there are no competing interests.

### Author Contributions

- Kattareeya Kumthip conceived and designed the experiments, performed the experiments, analyzed the data, prepared figures and/or tables, authored or reviewed drafts of the paper, and approved the final draft.
- Pattara Khamrin conceived and designed the experiments, performed the experiments, analyzed the data, prepared figures and/or tables, and approved the final draft.
- Hiroshi Ushijima conceived and designed the experiments, prepared figures and/or tables, and approved the final draft.
- Limin Chen and Shilin Li analyzed the data, prepared figures and/or tables, and approved the final draft.
- Niwat Maneekarn conceived and designed the experiments, authored or reviewed drafts of the paper, and approved the final draft.

### Human Ethics

The following information was supplied relating to ethical approvals (i.e., approving body and any reference numbers):

This work was conducted under the approval of the Research Ethics Committee of the Faculty of Medicine, Chiang Mai University (MIC-2557-02710)

### Data Availability

The nucleotide sequences of polymerase and capsid genes are available at GenBank: MN245671–MN245579 and MN253492–MN253541.

## Supplemental Information

Supplemental information for this article can be found online at http://dx.doi.org/10.7717/peerj.8520#supplemental-information.

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
