# Peer review of "Genetic recombination and diversity of sapovirus in pediatric patients with acute gastroenteritis in Thailand, 2010–2018"

_PeerJ, doi:10.7717/peerj.8520_

## Round 0.1 · original submission · Minor Revisions

I am happy to provisionally suggest that you manuscript be accepted for publication pending revision according to the suggestions made by the reviewers. All of the suggested changes represent minor updates to the manuscript, and are generally directed at providing additional details and clarification of methods, and inclusion of additional references.

Reviewer 1 ·

Basic reporting

The article by Kumthip and collaborators describes the prevalence of human sapovirus among children in Thailand for a period of 9 years. The importance of sapovirus is becoming increasingly recognize as a cause of viral AGE in people of all ages, therefore reports on its prevalence and genetic diversity are necessary. In that sense this manuscript is well timed. I found however that some aspects of the study hasn't been properly described or analyzed in the manuscript as it is in its current form. I will give some more detailed comments in the next sections.

Experimental design

The authors should include a definition of what acute gastroenteritis was considered in this study and also specify the age range of the individuals included in it.
in the end of the introduction I am missing a sentence in which the authors clearly defined which knowledge gap this study is filling.
In the sequence analysis and genotype identification paragraph ( lines 97 to 107), in particular in line 101the authors mentioned the use of one primer for sequencing, but I assume they used both ( forward and reverse), if so, please modify, otherwise please justify why not because sequencing techniques can have errors and by aligning and curating both sequences these errors can be fixed, if their sequences are based in only one read their accuracy can be compromised.
There is a recently released typing tool for calicivirus, which include sapovirus (https://norovirus.ng.philab.cdc.gov), I would suggest the authors to also try and type their sequences using it.
I noticed the authors didn't specify the type of phylogenetic tree and the model used, but they did mention it in the tree figure legends, please also do so in the material and methods section.
In the results the presence of confections is mentioned, and also in one of the tables, I would suggest the authors to consider adding this testing to the material and methods section.

Validity of the findings

I am missing some information regarding the location of the five hospitals, and if that locations can give a good overview of the sapovirus distribution in the whole country, or only of some particular regions. On that same idea, did the authors observed any particular genotypes in a particular location? or any seasonal trends? maybe include some information on this in the results and /or discussion sections.
In line 227 the authors suggest that sapovirus classification should be based on both genes: capsid as well as polymerase, but they do not give any justification for that statement, I would mentioned that this is the direction that norovirus classification is going, and since sapovirus is a very similar pathogen in many aspects, it only makes sense to do the same for sapovirus, maybe include a reference as well (J Gen Virol. 2019 Oct;100(10):1393-1406. doi: 10.1099/jgv.0.001318).
Also, in lines 229 and 230 the authors said that the results "revealed the significance of the novel recombinant virus", that is just an empty statement if they have a possible hypothesis on the importance of recombinants, or biological implications, please elaborate that idea.

Additional comments

In line 62 the authors mentioned 17 genotypes, and the additional GII.8, but currently GII.8 is a well accepted genotype, and they can confidently said that sapovirus is divided in 18 genotypes. Furthermore, an additional GII genotype ( provisionally named as GII.NA) has been already proposed (Near-Complete Human Sapovirus Genome Sequences from Kenya. 2019. Diez-Valcarce M, Montmayeur A, Tatusov R, Vinjé J.)
In table 3 I suggest the authors show the patients age in only one unit, probably month, for clarity.Also in that legend specify the highlighted samples are the ones in which recombinant sapovirus were found.

·

Basic reporting

The manscript describes a study on the prevalence and diversity of sapoviruses in pediatric patients in Thailand. In addition, evidences for existence of recombinant strains are presented.
The manuscript is well written using unambigous scientific language.
Some recent references on prevalences of sapoviruses in human patients from different geographic areas are missed, some of them indicxating higher prevalences than those usual reported. For instance: Bucardo et al. 2014 doi:10.1371/journal.pone.0098201; Varela et al., 2019 doi:10.3390/v11020144; Thongprachumet al. 2015, doi:10.1093/infdis/jit254; Biscaroet al. 2018, Infect. Genet. Evol. 2018, 60, 35–41, among others. A review of the recent literature is encouraged, since it would improve the discussion section.

Experimental design

The experimental design is correct and the research is within the scope of the journal. The objectives are well defined and the methods described with sufficient detail.

Validity of the findings

Conclusions are well based on the results obtained and linked to the original objectives.

Additional comments

In general, the manuscript is well written, the experimental design is correct and the results and conclusion can be of interest for the potential readers of the jurnal. A review of recent literature is encouraged, which can improve the discussion section.

---

## Round 0.2 · accepted · Accept

I am happy to accept your manuscript for publication. Congratulations!